

# Patterns of bleaching and mortality following widespread warming events in 2014 and 2015 at the Hanauma Bay Nature Preserve, Hawai'i

Ku'ulei S. Rodgers[*], Keisha D. Bahr[*], Paul L. Jokiel[†] and Angela Richards Donà[*]

Hawai'i Institute of Marine Biology, University of Hawai'i, Kāne'ohe, HI, United States of America
[*] These authors contributed equally to this work.
[†] Deceased.

Corresponding author
Angela Richards Donà,
angelard@hawaii.edu

## ABSTRACT

Drastic increases in global carbon emissions in the past century have led to elevated sea surface temperatures that negatively affect coral reef organisms. Worldwide coral bleaching-related mortality is increasing and data has shown even isolated and protected reefs are vulnerable to the effects of global climate change. In 2014 and 2015, coral reefs in the main Hawaiian Islands (MHI) suffered up to 90% bleaching, with higher than 50% subsequent mortality in some areas. The location and severity of bleaching and mortality was strongly influenced by the spatial and temporal patterns of elevated seawater temperatures. The main objective of this research was to understand the spatial extent of bleaching mortality in Hanauma Bay Nature Preserve (HBNP), O'ahu, Hawai'i to gain a baseline understanding of the physical processes that influence localized bleaching dynamics. Surveys at HBNP in October 2015 and January 2016 revealed extensive bleaching (47%) and high levels of coral mortality (9.8%). Bleaching was highly variable among the four HBNP sectors and ranged from a low of ∼31% in the central bay at Channel (CH) to a high of 57% in the area most frequented by visitors (Keyhole; KH). The highest levels of bleaching occurred in two sectors with different circulation patterns: KH experienced comparatively low circulation velocity and a low temperature increase while Witches Brew (WB) and Backdoors (BD) experienced higher circulation velocity and higher temperature increase. Cumulative mortality was highest at WB (5.0%) and at BD (2.9%) although WB circulation velocity is significantly higher. HBNP is minimally impacted by local factors that can lead to decline such as high fishing pressure or sedimentation although human use is high. Despite the lack of these influences, high coral mortality occurred. Visitor impacts are strikingly different in the two sectors that experienced the highest mortality evidenced by the differences in coral cover associated with visitor use however, coral mortality was similar. These results suggest that elevated temperature was more influential in coral bleaching and the associated mortality than high circulation or visitor use.

## INTRODUCTION

Global sea surface temperatures (SSTs) have increased an average 0.9 °C in the past century due to an increase in anthropogenic atmospheric gases resulting mainly from fossil fuel burning (*Sabine et al., 2004*). The greatest increases of 0.06–0.11 °C decade$^{-1}$ have occurred since 1970 (*EPA, 2016*) and have resulted in mass coral bleaching events worldwide. Scientific documentation of these events began nearly a decade later (*Jaap, 1979*). Since then, large-scale bleaching has occurred worldwide with increasing frequency and severity, and is projected to continue (*Hoeke et al., 2011*; *Mora et al., 2014*; *Bahr, Jokiel & Rodgers, 2015*). Nearly half of the corals in the western Indian Ocean were lost following widespread bleaching in 1998. By 2005, SSTs in the Caribbean had surpassed any previously reported temperatures and caused unprecedented coral mortality (*Eakin, Lough & Heron, 2009*). The year 2014 marked the beginning of the longest global bleaching event on record, which currently continues and has affected more reefs than any previous worldwide bleaching event (*Eakin et al., 2014*). Australia's Great Barrier Reef (GBR) recently experienced catastrophic bleaching and mortality with over 90% of its 2,300 km reef tract affected (*ARC, 2016*). The pristine reefs of the northern GBR were thought to be resistant to bleaching due to their remote location and low fishing and tourism pressure, however, over 99% of these reefs were observed bleached along a 1,000 km stretch (*Normile, 2016*). On Kiritimati atoll, over 80% mortality occurred during a record 15 months with SSTs above local bleaching thresholds. By November 2016, up to 90% of corals were dead (*Baum Lab, 2016*). This devastating loss of coral occurred on the relatively undisturbed reefs in the southeastern part of the atoll as well as in the degraded northwest (*Sandin et al., 2008*; *Watson, Claar & Baum, 2016*). The full extent of worldwide coral mortality has not yet been quantified, however, NOAA climate models predict another year of warming for the GBR, Kiritimati, and other Pacific Islands, particularly in the southern hemisphere (*NOAA Coral Reef Watch, 2017*). The negative influence of prolonged elevated seawater temperature on coral reefs is not selective and appears to affect protected, pristine, and degraded reefs, equally.

Coral reefs of the Hawaiian Islands have not been exempt, experiencing extensive bleaching in recent years. Significant heating in the offshore waters statewide (+1.15° over the past 58 years) has led to an increase in frequency of coral bleaching events (*Jokiel & Brown, 2004*; *Bahr, Jokiel & Rodgers, 2015*). The bleaching events that affected the Hawaiian Islands in 1996, 2002, and 2004 were relatively short in duration and thus coral recovery was high (*Jokiel & Brown, 2004*). Conversely, Hawaiian reefs experienced unsurpassed bleaching on a statewide scale during the multi-year bleaching events in 2014 and 2015 (*Bahr, Jokiel & Rodgers, 2015*).

The Hanauma Bay Nature Preserve (HBNP), a 40 ha fully protected Marine Life Conservation District (MLCD) established in 1967, is the most popular snorkeling location in the Hawaiian Islands with close to one million visitors annually (Fig. 1). The Coral Reef Assessment and Monitoring Program (CRAMP) began surveys here in 2000 and has shown a significant decline in shallow water coral cover with the majority of the decrease occurring after 2002 (*Brown et al., 2004*; *Rodgers et al., 2015*). The recent and predicted bleaching (*Jokiel & Brown, 2004*; *Hoeke et al., 2011*; *Eakin et al., 2014*) poses an imminent

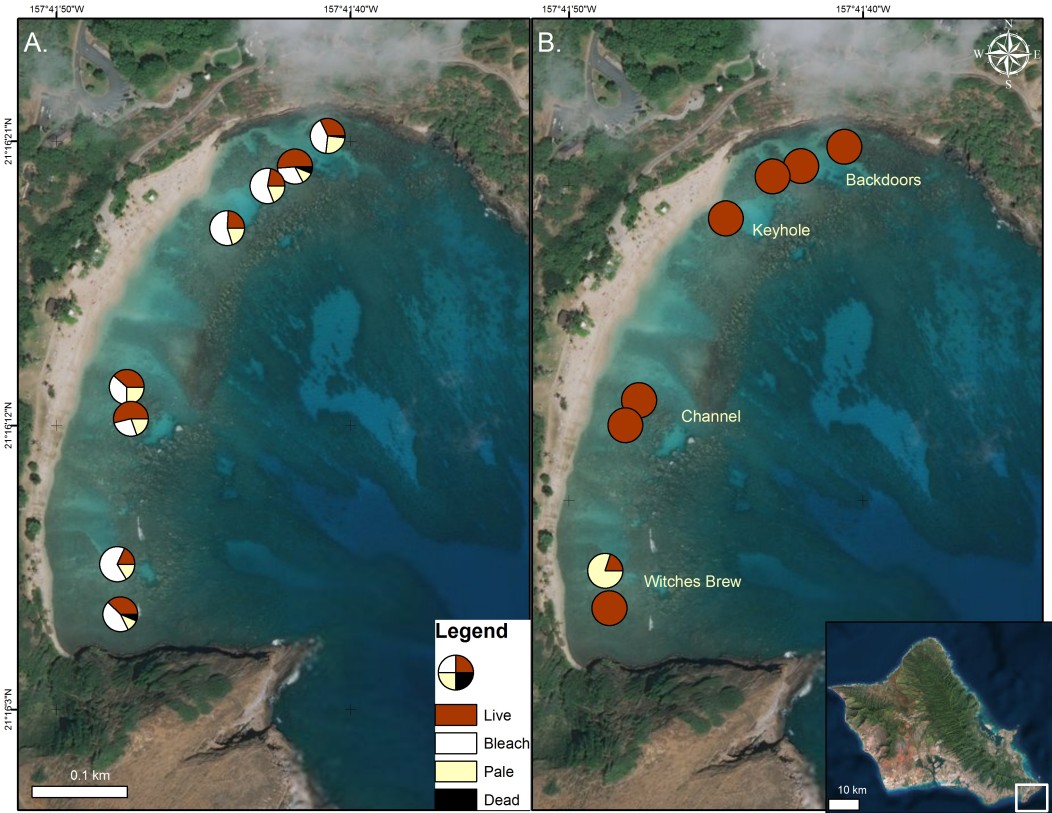

**Figure 1** **Coral condition on map of Hanauma Bay Nature Preserve.** Coral condition in the Hanauma Bay Nature Preserve, Oʻahu, Hawaiʻi in October 2015 (A) and January 2016. (B) Proportion of surveyed corals are shown as normal (red), bleached (white), pale (yellow), and dead (black). Surveys were conducted and temperature loggers deployed at each of the two stations within the four sectors. Photo credit: Quickbird Digital Globe.

threat to the biological sustainability of the HBNP ecosystem and a significant economic threat to the state of Hawaiʻi. Tourism expenditures provided over 15 billion USD to the state's economy in 2015 (*DBEDT, 2016*). Of the over eight million annual visitors to the state of Hawaiʻi, it is estimated that 80% participate in ocean recreational activities and over 1,000 ocean recreation companies exist to accommodate them (*Clark, 2016*).

The HBNP is located adjacent to the strong open-ocean, westward current referred to as the Molokaʻi Express. The outer section of this 40 ha (100 ac) bay can at times experience strong surges, ocean currents, and high wave energy while the protected 8 ha (20 ac) inner reef, located shoreward of the reef crest is relatively calm (*Brock & Kam, 2000*). The overall circulation pattern within Hanauma Bay moves shoreward and westerly from the northeast Toilet Bowl side of the bay towards the southwest Witches Brew side (Fig. 2). This pattern prevails during all tide phases (incoming, outgoing, and mixed) and during both tradewind and calmer south wind conditions. The mean velocity is 3.1 cm sec$^{-1}$ with a range of 0.8–6.5 cm sec$^{-1}$ with higher average velocities near the outer mouth of the bay and decreasing in shallower inner waters (*Whittle, 2003*). Under typical tradewind conditions water continuously enters the inner reef across the reef and boulder boundary.

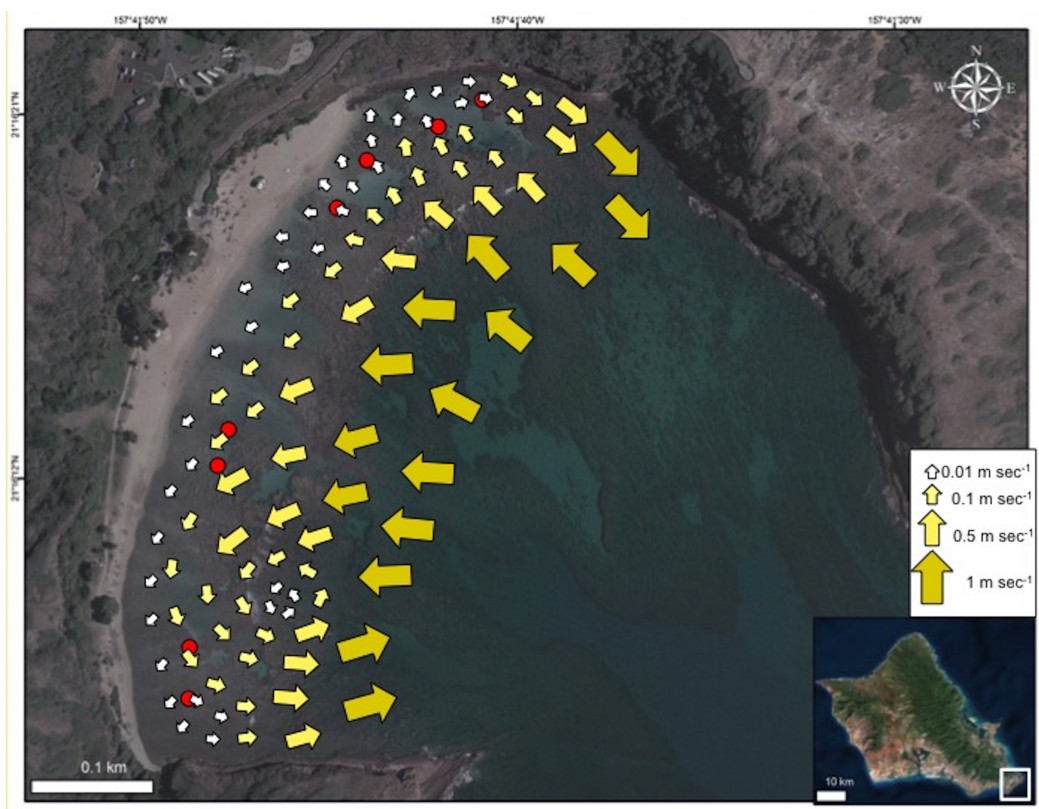

**Figure 2** **Map of current flow at Hanauma Bay Nature Preserve.** Generalized map of current flow in Hanauma Bay Nature Preserve, Oʻahu, Hawaiʻi. Red dots indicate surveyed sites. Photo credit: Quickbird Digital Globe.

Travelling parallel to shore it exits at high velocities (up to 50 cm sec$^{-1}$) near the ledges on the extreme opposite sides of the bay (Fig. 2) and through the mid-bay channel during outgoing tide at lower speeds.

In 2014, the State of Hawaiʻi Division of Aquatic Resources (DAR) coral bleaching assessments determined 47% of corals exhibited signs of bleaching in the HBNP; however, mortality was not subsequently quantified (*Neilson, 2014*). The main objective of this study was to quantify bleaching prevalence and subsequent mortality within the four major sectors of the HBNP and define how they relate to temperature and currents. These scientific data form a benchmark for the local environmental patterns that can be used to predict the extent and distribution of future bleaching events and aid management preparation strategies.

## METHODS & MATERIALS

### Coral surveys

In 2015, surveys were conducted to characterize coral bleaching extent and severity in the four major inshore sectors of the Hanauma Bay Nature Preserve (HBNP) (21.2690°N, 157.6938°W). These areas are locally known as Backdoors (BD), Keyhole (KH), Channel (CH), and Witches Brew (WB) (Fig. 1). To define the extent of bleaching, two 15 m × 5 m

transects were surveyed in each sector. Transect lines were placed on the reef flat at depths <1 m and all coral colonies within the transect area (75 m$^2$) were counted. To quantify the severity of the bleaching and mortality, we recorded coral species, colony size, and percent of colony that was live, pale, bleached, and recently dead. Redundant methodologies were used to provide accurate locations for subsequent resurveys using a handheld Garmin Geko 201 GPS unit, graphic and written documentation of positions using triangulation, and underwater photographic imagery of distinct initial and concluding coral colonies on each transect. To avoid error from observer variation, one surveyor collected data in all sectors during both initial bleaching and recovery surveys.

## Temperature

From June 2015 to January 2016, seawater temperature at all four sectors within the HBNP was recorded at fifteen-minute intervals using replicate HOBO Water Temperature Pro v2 Data Loggers (Onset, Wareham, MA, USA). The loggers were secured in 6″× 12″ hand-poured concrete "rocks" that mimic the benthic substrate and protect the loggers from solar irradiance and associated heating (*Bahr, Jokiel & Rodgers, 2016*) while providing concealment from human disturbance. The loggers at BD, KH, and CH were placed at 1 m depth, whereas at WB, a somewhat deeper site, the loggers were placed at 3 m to determine temperatures at a long-term monitoring station. To adjust for depth and to determine whether there were finer scale variations, an additional short-term deployment of 32 loggers were deployed at all transect locations on 19 April 2016 from 08:00 (low tide) to 15:00 (high tide). These data were used to calculate mean mid-day differences among transects.

## Currents

Current patterns characterized by *Whittle (2003)* primarily covered the region seaward of the reef boundary. To determine the nearshore current patterns, drogues were released and tracked in the water by surveyors. Twenty 2.5″ soft plastic balls were deployed and followed in each of the four sectors ($n = 80$). Drogues were identified by color and numbers written on 4″× 4″ underwater paper attached with a cable tie through small holes in the plastic. These holes also allowed the drogues to fill with water and remain positioned just below the water surface where wind effects are negligible. The position of each drogue was determined by GPS (Garmin eTrex 10) at initial deployment and at each subsequent sighting for eight consecutive hours spanning both low (−0.05 ft) and high (1.15 ft) tides. Drogues were deployed shoreward ($n = 40$) and seaward ($n = 40$) of the reef boundary. Researchers swam parallel to shore in a creeping line pattern from one end of the designated sector to the next dropping each drogue approximately 10 m apart horizontally with 5 m between each deployment line. After eight hours, drogues were retrieved and final location recorded. Drogues reaching shore prior to the end of the final retrieval were randomly redeployed in their original sector. To determine the original sector when drogues moved to another sector, each sector had different colored drogues.

Most of the drogues deployed at BD were carried outside the bay and were not followed by swimmers thus current patterns for this sector remained unclear. Additionally, nearshore areas in the CH sector were not fully covered. Thus, a second deployment was conducted

**Table 1  Coral condition by sector in October 2015 and January 2016.** Coral condition in October 2015 and January 2016 in the Hanauma Bay Nature Preserve, Oʻahu, Hawaiʻi (mean ± SE). Mean coral condition in 75 m² surveyed area.

| Sector | Live | Bleach | Pale | Dead |
|---|---|---|---|---|
| **October 2015** | | | | |
| **BD** | 37 ± 6.3 | 38 ± 6.2 | 22 ± 4.1 | 2.9 ± 1.7 |
| **CH** | 47 ± 6.4 | 31 ± 5.3 | 22 ± 4.6 | 0.3 ± 0.3 |
| **KH** | 23 ± 6.1 | 57 ± 7.1 | 20 ± 4 | 0.8 ± 0.6 |
| **WB** | 30 ± 5.2 | 53 ± 5.6 | 13 ± 2.9 | 3.9 ± 2 |
| **January 2016** | | | | |
| **BD** | 94 ± 3 | 0.5 ± 0.4 | 0.7 ± 0.7 | 0 |
| **CH** | 77 ± 4.8 | 0 | 0.2 ± 0.2 | 0.5 ± 0.5 |
| **KH** | 85 ± 4.6 | 0 | 0 | 0.3 ± 0.3 |
| **WB** | 85 ± 3.6 | 8 ± 3.1 | 2.5 ± 1.7 | 1.1 ± 0.5 |

during an incoming tide beginning at low tide (−0.23 ft). In the smaller BD sector, five drogues were released approximately 5 m apart along the eastern boundary. These were deployed over the reef in a perpendicular path to shore from the channel marker buoys. In the larger CH sector, five drogues were released approximately 10 m apart following a path parallel to shore. All drogues were deployed shoreward of the reef boundary and retrieved after three hours with final locations recorded.

### Statistical analyses

Bleaching prevalence was analyzed using a General Linear Model (GLM) with sector as a fixed factor and transect nested within sector. Temperature was treated with a repeated measures mixed model by location with transect nested within location. Assumptions of normal distribution, homoscedasticity, and multivariate normality were assessed through graphical analyses of the residuals. All statistical analyses and descriptive statistics were conducted using JMP Pro 12. Calculations of location, distance, and time were determined in ArcGIS 10 and Excel 2010 to characterize current patterns (Fig. 2).

## RESULTS

### Bleaching prevalence

In October 2015, 45% ± 3.2% (mean ± SE) of corals in the Hanauma Bay Nature Preserve (HBNP) showed signs of bleaching (Table 1). The highest bleaching prevalence was observed in *Pavona varians* and *Pocillopora meandrina* (Fig. 3). Bleaching prevalence was significantly different among sectors (GLM; $F_{(7,143)} = 3.4239$ $p = 0.0020$) with highest levels at Keyhole (KH; 56.6 ± 7.1%) and Witches Brew (WB; 52.7 ± 5.6%) compared to Backdoors (BD; 38.4 ± 6.2%) and Channel (CH; 30.9 ± 5.3%) (Fig. 1). A further 13–22% of corals were paling in all sectors. Coral colony size was not a factor in bleaching prevalence ($R^2 = 0.0246; p = 0.0611$) whereas number of colonies was. While colony size in all locations was similar (Oneway ANOVA $F_{(3,290)} = 0.7229$, $p = 0.5391$), number of colonies at WB was higher (28 ± 2.9) (Oneway ANOVA; $F_{(3,12)} = 7.4677$ p<0.0044) compared to the average number of colonies at BD (15.25 ± 2.65), CH (19 ± 1.8), and KH (11 ± 0.8).

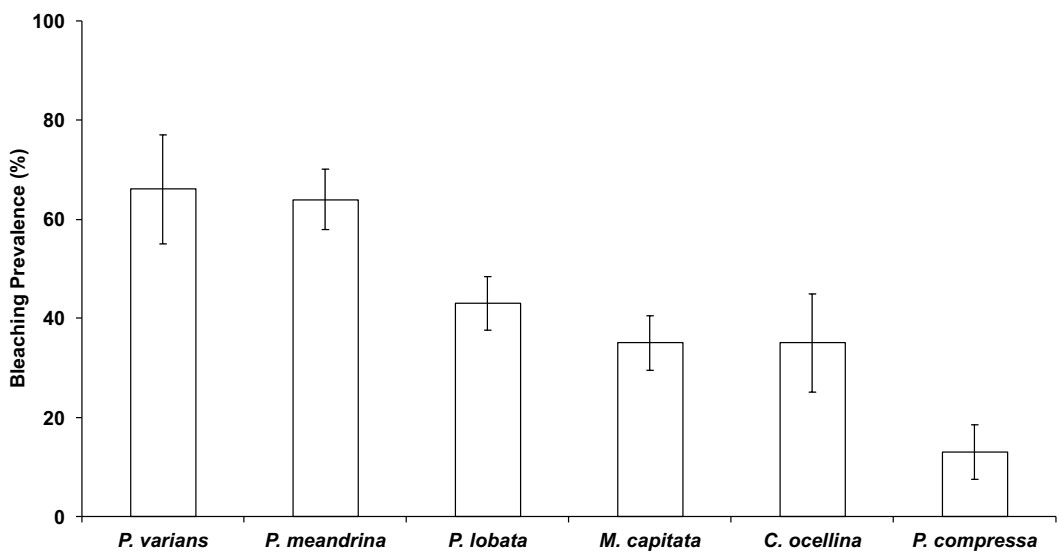

**Figure 3  Bleaching prevalence by coral species.** Mean bleaching prevalence by coral species in October 2015 in Hanauma Bay Nature Preserve, Oʻahu, Hawaiʻi. Error bars represent standard error.

## Coral mortality

In October 2015, the highest average levels of coral mortality occurred at WB ($3.9 \pm 2\%$) and BD ($2.9 \pm 1.7\%$). Lower mortality rates were observed at KH ($0.8 \pm 0.6\%$) and CH ($0.3 \pm 0.3\%$). In January 2016, average coral mortality rates by sector were near or below 1% (WB = $1.1 \pm 0.5\%$; BD = 0%; KH = $0.3 \pm 0.3\%$; and CH = $0.5 \pm 0.5\%$), although recovery was slowest in WB. Highest mortality rates were observed in *Porites lobata* and *Pocillopora meandrina* (Fig. 4). Total coral mortality inside the HBNP due to elevated SSTs was calculated with a cumulative value of 9.8% (October = 7.9%; January = 1.9%). Cumulative mortality rates also varied by sector (sum of Oct and January surveys: WB = 5.0%; BD = 2.9%; KH = 1.1%; and CH = 0.8%). No overlap in recently dead corals occurred between October and January determined by rapid algal turf growth over the coral skeleton.

## Environmental drivers

The patterns of coral bleaching prevalence and mortality in the four sectors in HBNP are linked to localized heating, due to circulation patterns. Incoming currents show a great reduction in flow velocity as oceanic water (1 m s$^{-1}$) flows over the reef boundary into the shallow, sandy areas (0.5 m s$^{-1}$). Here, residence time and temperatures increase and the warmer water follows a westerly direction (0.1 m s$^{-1}$) along shore to the far end (WB) where it turns seaward and flows either to a small gyre or over the reef boundary and out of the bay (Fig. 2). Because water exits the bay at WB, water flow into the sector comes strictly from slow, alongshore currents. A similar small gyre is located at the far eastern end (BD), whereas at the adjacent KH, water flows slowly to shore at 0.01 m s$^{-1}$ (Fig. 2). Inshore currents were faster in CH ($0.075 \pm 0.011$ m s$^{-1}$) and WB ($0.055 \pm 0.011$ m s$^{-1}$) compared to KH ($0.008 \pm 0.011$ m s$^{-1}$) and BD ($0.019 \pm 0.011$ m s$^{-1}$) (One Way ANOVA; $F_{(3,7)} = 7.717$; $p = 0.0386$). Notably, the relatively high current velocities at CH and WB
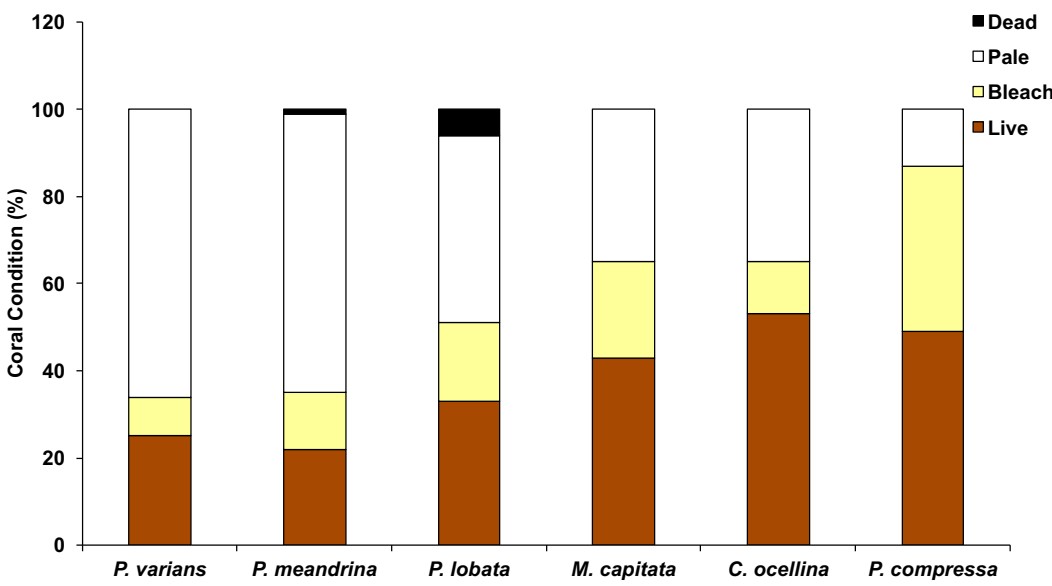

**Figure 4** **Graph of coral condition in October 2015.** Coral condition in October 2015 in Hanauma Bay Nature Preserve, Oʻahu, Hawaiʻi across all sectors. Mean proportion of surveyed corals are indicated by color (Live, brown; pale, yellow; bleach, white; and black, dead).

are substantially different in character. At CH cooler water flows into the shallows from outside the reef boundary, whereas at WB, slow-moving, warm water from as far as KH sector increases in velocity only as it is exiting the shallows.

Analysis of temperature gradients in HBNP revealed temperatures were significantly different throughout the bay (Mixed Model; $R^2 = 0.30$; $F_{(3,4)} = 454.97$; $p < 0.0001$). WB experienced significantly higher mean mid-day temperatures ($26.46 \pm 0.014\,°C$) compared to BD ($26.05 \pm 0.014\,°C$), CH ($26.01 \pm 0.014\,°C$), and KH ($25.99 \pm 0.014\,°C$). Additionally, temperatures are variable within locations (Mixed Model; $R^2 = 0.43$; $F_{(1,2)} = 23.27$; $p < 0.0001$). Overall, the largest differences within site occurred at WB ($\pm 0.58\,°C$) and BD ($\pm 0.26\,°C$), while KH ($\pm 0.10\,°C$) and CH ($\pm 0.08\,°C$) were more similar. The effects of the localized heating revealed higher bleaching prevalence in KH ($56.6 \pm 7.1\%$) and WB ($52.7 \pm 5.6\%$) compared to BD ($38.4 \pm 6.2\%$) and CH ($30.9 \pm 5.3\%$) (Fig. 1).

## DISCUSSION

Bleaching and recovery rates and species tolerance in 2015 were highly variable across islands of the Hawaiian archipelago. In 2015, the warm water approached the islands from the south and this resulted in a maximum 18 DHW for Hawaiʻi Island and 10 DHWs for eastern shores of Oʻahu (*NOAA Coral Reef Watch, 2016*). Extensive surveys in 2015 found between 30 and 86% bleaching on Hawaiʻi Island with reported mortality at nearly 50% on the island's west coast (*Kramer et al., 2016*). Results of the current study revealed nearly half (47%) the corals surveyed at HBNP were found to exhibit signs of severe bleaching and associated mortality was ∼9.8%. It appears localized heating and circulation patterns inside HBNP are driving differences in bleaching-associated mortality. Variation in spatial and temporal temperature patterns account for the differences in bleaching between islands.
Accompanying rates of coral mortality may slow predicted recovery rates of Hawaiian corals and shape future reefs (*Jokiel & Brown, 2004*; *Bahr, Jokiel & Rodgers, 2015*).

The Hanauma Bay Nature Preserve (HBNP) has shown decline in coral cover in shallow waters since 2002 (*Brown et al., 2004*; *Rodgers et al., 2015*) however, global climate change may drive this management-protected reef into more rapid decline. Increasing length, severity, and frequency of coral bleaching events pose an imminent threat to the biological sustainability of the HBNP ecosystem and a significant economic threat to the state of Hawai'i. Of the total area surveyed (600 m$^2$) in the HBNP in 2015, cumulative coral mortality was 9.8%. This Marine Life Conservation District reflects the fish populations in more remote areas distant from anthropogenic impacts due to management restrictions that prohibit any take of marine organisms. However, the organic and nutrient levels at HBNP are much higher than at 60 other sites statewide due to high fish biomass (*Rodgers, 2005*). Minimal levels of fine sediments due to a low contribution of terrigenous material from the influencing watershed are found here. In addition, nearly one million people visit HBNP annually but the majority of visitors using the ocean resources remain on the northern end of the bay in the BD and KH regions. The southern section where WB lies has relatively minimal human use (Fig. 1). Nonetheless, these two areas experienced similar mortality following the bleaching event. These results support global reports of high mortality following bleaching in remote regions removed from anthropogenic influences such as the northern GBR and at Lisianski in Papahānaumokuakea in the Northwestern Hawaiian Islands (NWHI) (*Couch et al., 2016*).

Reef recovery after major disturbances depends not only on the prevailing environmental conditions but also on the species affected. For example, *Pocillopora meandrina* is considered a "competitive" species (*Darling et al., 2012*) and is far more likely to recolonize a degraded reef than longer-lived "stress-tolerant" species such as *Porites lobata and Porites evermanni*. This study revealed bleaching prevalence and mortality to vary by species and location. The highest bleaching prevalence was observed in *Pavona varians* (66%), and *P. meandrina* (64%) while highest mortality occurred in *P. lobata* (5.3%) and *P. meandrina* (1.3%). No mortality was observed in *P. varians*. Bleaching prevalence was highly variable within HBNP due to localized environmental gradients. The highest levels of bleaching and mortality were observed in WB, which is characterized as the sector where warm water accumulates before exiting the bay.

Additionally, WB has the greatest number of coral colonies, particularly *Porites lobata* colonies. *Porites lobata* was also the most abundant species at BD. This sector is characterized as having low water velocity and relatively high temperatures. Cumulative bleaching was relatively low compared to other sectors but mortality was second highest. KH has the lowest number of coral colonies and *P. meandrina* was the most abundant species. Currents head directly from beyond the reef boundary to shore presumably bringing colder water with considerable reduction in velocity into KH. Temperature is lower here than at WB and BD, which may explain why mortality was low although cumulative bleaching was high. Lastly, CH had the lowest bleaching and mortality due to high water circulation and high oceanic input with associated lower temperatures. This sector had the second highest coral abundance, mainly dominated by *P. lobata* and *Montipora capitata*. Because *P. lobata*

is the most abundant species in three of four sectors, the observed mortality indicates an important vulnerability that cannot be overcome by circulation or conservation effort. With repeated mortality of the more vulnerable species, shifts in coral composition are likely to occur.

Temperature and circulation are difficult to separate. These two factors are highly correlated with one another since circulation can increase or ameliorate temperatures and account for localized heating differences. Our results suggest circulation patterns facilitate localized heating and influence bleaching dynamics in HBNP. Incoming oceanic water flows shoreward to the reef boundary then follows a counterclockwise pattern west before exiting the bay. Significant heating occurs as incoming water moves over the shallow reef flat and accumulates in the WB sector. This explains the observed ∼0.5 °C difference in temperatures between WB and other sectors of HBNP. The observed high circulation rates appear to be movement and accumulation of warm water in WB. This accumulation of warm water likely facilitated increased bleaching prevalence and associated mortality in that area. A similar, but less severe pattern was observed in the BD sector (Fig. 2). The cumulative heating associated with these circulation patterns correlates with the observed high levels of bleaching in these two sectors. Even in the absence of direct anthropogenic stressors (e.g., fishing pressure, pollution, and sedimentation) coral mortality can be high as temperatures increase. Corals live within 1–2 °C of their summer maximum temperatures and will bleach at this threshold whether they inhabit cooler, deeper waters or live on warmer shallow reefs (*Coles, Jokiel & Lewis, 1976*).

## CONCLUSION

In summary, bleaching and mortality were highly variable across the main Hawaiian Islands. Differing spatial patterns of warming greatly influence the location and severity of bleaching and associated mortality. Results of this study indicate variability in bleaching and associated mortality can be described by species-specific tolerances, number of colonies, localized environmental patterns of heating, and currents. This study of the marine protected HBNP results determined:

- Bleaching and mortality varied by species and location.
- Bleaching prevalence and associated mortality were the highest in the sectors where warm water accumulated (i.e., BD and WB).
- Regardless of anthropogenic influences, temperatures beyond the thermal tolerances for corals can result in mortality.

Oceans will continue to absorb a significant amount of carbon even once emissions are reduced but we must slow the increase to begin addressing the impacts of climate change. Sound management strategies based on scientific research will increasingly play a more important role. Data from this research will serve as a baseline for future research to better understand the environmental patterns in HBNP and elsewhere. Data on species tolerances, circulation patterns and temperatures can assist managers in predicting the spatial extent, bleaching severity, and distribution of future bleaching events to support planning efforts.

## ACKNOWLEDGEMENTS

Drogue tracking and temperature logger placement was accomplished with the assistance of members of the Coral Reef Ecology Lab at the Hawaii Institute of Marine Biology (Yuko Stender and Megan Onuma). Access to the Hanauma Bay Nature Preserve was granted through Supervisor, Tara Hirohata, and Ocean Recreation Specialist, Kaipo Perez III Ph.D. The authors also recognize the guidance and historical bleaching research conducted by Paul L. Jokiel. This is the Hawai'i Institute of Marine Biology (HIMB) contribution #1684 and the School of Ocean and Earth Science and Technology (SOEST) contribution #10004.

### Funding

The authors received no funding for this work.

### Competing Interests

The authors declare there are no competing interests.

### Author Contributions

- Ku'ulei S. Rodgers conceived and designed the experiments, performed the experiments, contributed reagents/materials/analysis tools, wrote the paper, prepared figures and/or tables, reviewed drafts of the paper.
- Keisha D. Bahr conceived and designed the experiments, performed the experiments, analyzed the data, wrote the paper, prepared figures and/or tables, reviewed drafts of the paper.
- Paul L. Jokiel conceived and designed the experiments, performed the experiments, contributed reagents/materials/analysis tools, provided historical bleaching research.
- Angela Richards Donà performed the experiments, wrote the paper, reviewed drafts of the paper.

### Data Availability

The raw data was provided as Data S1.

### Supplemental Information

Supplemental information for this article can be found online at http://dx.doi.org/10.7717/peerj.3355#supplemental-information.

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
