# Peer review of "Patterns of bleaching and mortality following widespread warming events in 2014 and 2015 at the Hanauma Bay Nature Preserve, Hawai‘i"

_PeerJ, doi:10.7717/peerj.3355_

## Round 0.1 · original submission · Minor Revisions

The reviewers have provided a good number of suggestions that would improve the interpretations of the results in the discussion. I would encourage the authors to address these comments/suggestions in a new revised version.

Reviewer 1 ·

Basic reporting

The basic reporting is good. More details on survey metrics would be helpful. Structure is professional and appropriate.

Experimental design

Experimental design was well planned and appropriate for the goals of this study.

Validity of the findings

Findings are valid, but the discussion should be more focused on the findings at this site and spend less effort discussion bleaching at larger scales. More discussion about the relationship between bleaching, temperature, and water motion at this location would improve the paper.

Additional comments

Paper:

Patterns of bleaching and mortality following widespread warming events in 2014 and 2015 at the Hanauma Bay Nature Preserve, Hawaii

Synopsis:

This paper describes an extensive survey of coral bleaching and mortality at the Hanauma Bay Nature Preserve associated with the 2014-15 bleaching events in Hawaii. The authors examine the bleaching and mortality in association with temperature and water motion. Water temperature was a significant driver of bleaching at this location. This paper provides useful information pertaining to the bleaching events occurring in Hawaii, and offers a sound experimental approach for examining the effects of environmental parameters on bleaching intensity. More details on the metrics of severity and extent would be helpful. Furthermore, it may be good to town down some of the statements regarding temperature overriding all other impacts on coral mortality, as only two environmental parameters were assessed in this study. Overall this is a great paper, and will contribute to the field of coral ecology and future studies of coral bleaching.


Abstract:

Line 10-11: I suggest changing to “…is increasing and data has shown isolated and protected reefs to be vulnerable to the effects of global climate change.”

Lines 13-14: I suggest rewording to “The location and severity of bleaching was strongly influenced by the spatial and temporal patterns of elevated seawater temperatures.”

Line 16: Change to “…HNBP, Oahu, to gain a…”

Lines 23: The sites names for WB and BD are not given prior to the use of acronyms here.

Line 31: Change to “These results suggest that…”

Lines 33-35: I suggest toning down this sentence. While elevated seawater temperature may override efforts, there are lots of studies showing how numerous environmental stressors exacerbate bleaching. Since there is no data on the threshold at which temperature overrides all protection efforts, this may be a stretch to state outright.


Introduction:

Lines 50-51: add ‘reef tract’ after 2,300km

Lines 51-52. Be more specific about the impacts. This statement is vague, and leads the reader to wonder ‘how devastating?’ Put the impact in context of the region and state the percentage of bleaching and mortality in these areas compared to those affected by human impacts.

Line 53: I suggest rewording to “…Islands have experienced extensive…”

Line 55: I suggest changing ‘facilitated’ to ‘led to’

Line 72: I suggest rewording to “…to the strong open-ocean West current, referred to as the Molokai Express.”

Lines 86-87: I suggest rewording to “…(DAR) coral bleaching assessments determined…”

Line 88: Change ‘research’ to ‘study’

Materials and procedures:

Line 96: It would be helpful to clearly define extent and severity, so it is clear what each parameter is quantifying, and how they were enumerated in the field.

Line 100: Add “the” between ‘in’ and ‘surveys’

Line 148: Add info pertaining to the statistical approaches. How did you ensure assumptions were met for the models? Were residuals examined? Any thresholds used? Were any transformations made?

Results:

Line 160: I suggest stating that prevalence was ‘significantly different’ among sectors rather than ‘highly variable.’ Use terms ‘significant’ throughout this paragraph to describe statistically significant findings.

Discussion:

Line 208: Again, define these terms clearly so the reader knows the definition of severity in the context of this paper.

Line 223-224: Tone down this statement and be specific to this study. Only two environmental parameters were measured, so it is more appropriate to state that temperature played a larger role than other parameters at this site. It also appears as those circulation and temperature were interconnected, so it may be more interesting to focus on the dynamics seen between those parameters.

Line 249-252: Please clarify this statement. It is clear that you are making the point that heating is more important than direct anthropogenic stressors, but please state the connection between the two measured parameters more explicitly. Based on your data, what appears to be the relationship between water motion and temperature? What are the primary patterns exhibited by the data?

Line 256: I’m not sure if this section is needed. It seems strange to end the discussion talking about other studies. Perhaps you could synthesize this into a shorter statement about regional bleaching and move to the conclusion. Then you could really provide a clear synthesized summary of your results at the end of the discussion, which are specific to the site used for this study.

Line 283: Again, severity and extent need to be more clearly defined early in the paper.

Line 295: As suggested above, integrate the info about bleaching in other regions into this section.

Reviewer 2 ·

Basic reporting

I like the manuscript a lot. It is clear and concise.

The methods are appropriate and are described well. Same with the results.

I would like much more discussion on why bleaching associated mortality was so variable across the region.

The authors use other data from the 2014-2015 event around the main Hawai'ian Islands to show the catastrophic nature of bleaching related mortality. However, the sites in their study on Oahu show much less morality in comparison. I am left wanting more -- more analysis and proposed hypotheses on why this is so.

I still believe the take home message is the same - but a few naysayers will argue that the mortality at HBNP does not necessarily "... support building evidence that elevated temperature undermines all ameliorating factors and is the most influential driver of coral bleaching and associated mortality"

Experimental design

I am fully satisfied with their design and methods

Validity of the findings

I believe the findings are significant but as I stated earlier I would like much more discussion on why coral mortality was so variable - why is this important? - and does it change the long view of the resilience of reefs in the face of warming SSTs? Is it just species specificity? Water depth? Circulation? DHWs? A combination of all of the above? Something else?

Additional comments

Please expand discussion and parse out the regional differences in mortality - why they are important and what do these data portend for reefs in the main Hawai'ian Islands

I commend the team for an outstanding study.

This ms to me is another brick in the wall - confirming that global climate change in all of its manifestations are trumping most if not all local impacts.

---

## Round 0.2 · accepted · Accept

The authors have addressed satisfactorily all the comments and suggestions from the reviewers.

Reviewer 1 ·

Basic reporting

I am happy with the revisions, and believe the manuscript is suitable for publication.

Experimental design

The authors did a good job providing more information on the metrics used and the statistical approach.

Validity of the findings

I appreciate the clarity and transparent discussion of the findings, and believe the manuscript has been improved from the first revision and is suitable for publication.